# Probabilistic Risk Assessment of Dietary Exposure to Cadmium in Residents of Guangzhou, China—Young Children Potentially at a Health Risk

**DOI:** 10.3390/ijerph19159572

**Published:** 2022-08-04

**Authors:** Florence Mhungu, Kuncai Chen, Yanyan Wang, Yufei Liu, Yuhua Zhang, Xinhong Pan, Yanfang Cheng, Yungang Liu, Weiwei Zhang

**Affiliations:** 1Guangdong Provincial Key Laboratory of Tropical Disease Research, Department of Toxicology, School of Public Health, Southern Medical University, Guangzhou 510515, China; 2Guangzhou Center for Disease Control and Prevention, Guangzhou 510440, China; 3Institute of Public Health, Guangzhou Medical University, Guangzhou 510440, China

**Keywords:** cadmium, dietary exposure, @RISK, margin of exposure (MOE)

## Abstract

Cadmium (Cd) and its compounds are hazardous environmental pollutants with renal toxicity and human carcinogenicity, with ingestion of contaminated foods representing the major mode of exposure. There have been a number of reports evaluating the Cd content in various foods; however, regarding the actual risk posed by dietary cadmium exposure, only a few reports are available in which single point evaluation (less accurate than multiple point evaluation) was employed. In this study, we used a margin of exposure (MOE) model and @RISK software (for multiple evaluation) to evaluate Cd-related health risk in the local Guangzhou residents at varying ages, through a comparison between the estimated monthly exposures and the provisional tolerable monthly intake (0.025 mg/kg body weight (b.w.)), based on the Cd contents in various food categories available locally (a total of 3964 food samples were collected from each of the 13 districts of Guangzhou between 2015 and 2019), which were determined by using inductively coupled plasma mass spectrometry. In this study, Cd was detected in 69.6% of the samples (averaged 0.120 mg/kg), and rice and its products, leafy vegetables, bivalves, and shrimp and crabs contributed most to Cd exposure (8.63, 3.18, 2.79, and 1.48 ng/kg b.w./day, respectively). The MOE values demonstrated the following tendency: the younger age group, the lower MOE, and its 95% confidence range for the (youngest) 3~6 year old group started from 0.92, indicating a health risk of young children, while that for the other age groups were all above 1.0. Our preliminary findings warrant further clarification using biomarker assays in the relevant population.

## 1. Introduction

Cadmium (Cd) is a metallic component which occurs naturally in the earth’s crust, while possessing extensive exposure and persistent toxicity in the environment [1]. Cd can also be discharged to the environment through anthropogenic activities and emissions since it is used during the production of several consumer and industrial materials such as microelectronics, plastic stabilisers and dyes. Industrialised procedures such as mining, mining filtering, and fertiliser production from phosphate ores are significant sources of Cd pollution [2,3].

Cd as a highly toxic heavy metal of substantial toxicity may selectively damage the kidneys [1], and chronic exposure to low-level Cd is associated with a number of health outcomes, such as renal failure, early onset of diabetes [4,5], osteoporosis, disrupted blood pressure regulation, increased cancer risk [6,7] and other diseases. Additionally, there is sufficient evidence that Cd compounds, such as chloride, oxide, sulfate, and sulfide, are carcinogenic in animals [5,8], and IARC has classified Cd as a proven human carcinogen (supported by both sufficient animal carcinogenicity and human epidemiological evidence) [9,10]. Many health agencies have therefore precautionarily established exposure standards for protecting people from excessive Cd exposure, yet the potential effects of long-term, low level Cd exposure on the health of residents are extremely diverse in susceptibility to intoxication.

Far more important than smoking and other factors, gestational exposure is the dominant source of Cd exposure in the general population of humans [11,12]. In 2011, at its seventy-third meeting, the Joint FAO/WHO Expert Committee on Food Additives (JECFA) instituted a provisional tolerable monthly intake (PTMI) of 25 μg/kg b.w. for Cd, considering the extended half-life of Cd in humans. A global estimate of mean dietary exposure to Cd from all foods for adults ranged from 2.2 to 12 μg/kg b.w. per month, or 9–48% of the PTMI, specifically at 11.9 μg/kg b.w. per month or 47% of the PTMI for European children. For adults from Europe, USA and Lebanon, dietary exposures to Cd ranged from 6.9 to 12.1 μg/kg b.w. per month (28–48% of the PTMI), particularly from 20.4 to 22.0 μg/kg b.w. per month (82–88% of the PTMI) for children aged 0.5–12 years from the USA and Australia [13]. As for Cd exposure in China, the total dietary exposure at a country level is 24 µg/kg b.w. per month in children aged 4–11 years (96% of the PTMI). These estimates were based on a one-day 24 h dietary recall, which might be less accurate/reliable than the 3 day 24 h estimate. More importantly, most previous reports dwelt more on the detection of Cd in foods, instead of combining the food contamination and consumption data for further health risk characterization [13]. Considering the non-proportional intake of foods per unit of bodyweight in children at varying ages, we hypothesised that human bodies at different stages of life may have a varied level of health risk associated with dietary Cd exposure.

In the present study, based on the determination of Cd contents in various foods sampled locally from Guangzhou, China, and the established dietary structures of the local populations, we made use of the provisional tolerable monthly intake (PTMI) for Cd, i.e., 0.025 mg/kg b.w. (established by the World Health Organisation [14]) as a threshold of health risk of Cd exposure and a parameter termed the margin of exposure (MOE), which is a newly occurring and more reliable and accurate model for the characterization of risk, to estimate the health risk of dietary Cd exposure in Guangzhou citizens at various ages. Interestingly, in recent decades, the MOE parameter has been commonly applied in environmental risk assessment, such as by the United States Environmental Protection Agency (USEPA) and the European Food Safety Authority (EFSA), especially for comparative risk assessments of carcinogens in alcoholic beverages, tobacco, cannabis and other illicit drugs [15,16,17].

## 2. Materials and Methods

### 2.1. Food Categories

In this study, a total of 3964 food samples were procured from retail shops that covered 12 districts of Guangzhou City during 2015 to 2019. Three streets from each district were sampled randomly as the monitoring sites. Retailer selection included supermarkets, agricultural wholesale markets, shops, and restaurants. Food categories included the following: grains (rice and rice products, maize) noodles, steamed bread, and nuts; live aquatic products (fish, shrimp, crabs, and bivalves); livestock meat (beef, lamb, pork and pork offal); poultry, egg and egg products; and vegetables (leafy, root, and bulb), fresh fruits, melons, fresh beans, soy products, vegetable oil, and edible fungus. Samples were submitted to the lab in original and sealed conditions.

### 2.2. Determination of Cd with Inductively Coupled Plasma Mass Spectrometry

To determine Cd levels in foods, minor adjustments were made to the formerly established method [18,19] with minor adjustments. About 200–500 mg of each sample was placed in a digestion tube, each containing a mixture of 5 mL of 98% nitric acid and 2 mL of 30% hydrogen peroxide. Concurrently, control tests were also prepared, and all mixtures were held for 1.5–2.0 h. Inductively coupled plasma mass spectrometry was used to determine Cd content. Cd was detected by Agilent 7700 ICP_MS (Agilent Technologies, Santa Clara, CA, USA), with a plasma gas flow of 15 L/min, a frequency power of 1500 watts, and an auxiliary gas flow of 0.40 L/min.

### 2.3. Quality Control for Cd Determination and Limit of Detection

For quality control of the determination, all Cd reference materials were qualified as heavy metals (certified by the Chinese scientific community). Nitric acid and hydrogen peroxide used in this study were of ultrapure grade. Ultrapure water (with a resistivity of 18.2 M Ω cm) was produced by a Millipore system (Kurita Water Industries, Carrollton, TX, USA), and purified water was purchased from Wahaha Corp (Hangzhou, China). National first-level standard material (GBW10035) Cd, Cd standard in wheat powder (purchased from Chinese Academy of Geographical Sciences, Beijing, China), 74 ± 3 µg × kg^−1^ for Cd, was used as a reference (quality control) for the detection of Cd in the samples. The limit of detection (LOD) for elemental Cd was 0.0006 mg/kg, and the limit of quantification was 0.02 mg/kg, with an accuracy of <5% deviation from the real values.

### 2.4. Estimation of Daily Food Intake

Statistics on food intake were obtained from the 2011 Chinese National Food Consumption Survey of Urban and Rural Residents. Dietetic intake material was hinged on a 3-day/24 h recall survey combined with an edible oil weighing method [20,21]. The process entailed surveying the habitual intake of 2960 residents from 998 households, including 1416 men and 1544 women. This period of dietary structure survey was longer than the ordinarily applied 24 h (1-day) protocol; for example, in the USA, the 24 h dietary recall protocol was routinely used in many rounds of the National Health and Nutrition Examination Study [22]. We preferred using a more accurate calculation of the typical intake and dietary structure in the local population by employing 3-day/24 h recall data, by which more dietary options may be recalled thus the measurement error that could affect survey findings and conclusions could be lowered [23].

Inner-city residents represented 63.8% of the total population, and suburban residents accounted for the other 36.2%. Participants ranged in age from 3 to 86 years, with an average age of 32 years. The 3 to 6, 7 to 17, 18 to 59, and 60 years and above age groups translated to 6.7, 21.5, 58.6, and 13.2% of the entire study population, respectively [24,25,26,27].

### 2.5. Estimation of Daily Intake of Cd

The average monthly exposure to Cd is measured as the average amount of food consumed per kilogram of body weight (B.W.) based on the number of days recorded, multiplied by the average concentration of Cd in each food group. The estimated monthly exposures were compared with the provisional tolerable monthly intake (PTMI) 0.025 mg/kg b.w. (equivalent to 25 μg/kg b.w.), as established by the Joint Food and Agriculture Organization of the United Nations/World Health Organisation Expert Committee on Food Additives [28].

The estimated daily intake (*EDI*) of Cd from total diet was calculated by using Equation (1).
(1)$$EDI=\overset{n}{\underset{i=1}{\sum }}=\frac{Di\times Mi}{W}$$

*EDI* is the estimation of daily intake of dietary Cd (ng/kg b.w./day). *Di* represents the daily intake of each food in each age group (g/person day^−1^). Mi denotes the mean level of Cd in each food category (μg/kg).When Cd was not detected in certain types of food, *Mi* was assumed to be LOD/2 [29]. *W* signifies the body weight of each age group (kg). The average weight of respondents at 3 to 6 years old was determined to be 20 kg [30], 40 kg [31] for respondents at 7 to 17 years, and 60 kg [32] for participants of the other two age groups, 18 to 59 years and 60 years and older.

### 2.6. Risk Classification

The study employed the Margin of Exposure (MOE) element to describe the risk characterization which is recommended by European Food Agency, Food and Agriculture Organisation and WHO. MOE is the gap between the actual intake of a substance by a given population and the estimated daily dose over a lifetime that experts consider to be safe [33,34]. Because MOE is the ratio between a stated point for the detrimental effect and human consumption, it makes no implicit assumptions about a “safe” intake, as a result, the Scientific Committee believes that this approach is more appropriate for chemicals that are both genotoxic and carcinogenic [35,36]. Additionally, the MOE technique is designed to provide risk managers with an idea of the level of alarm and to aid in determining the necessity and immediacy of further action. Additionally, varied estimations of daily nutrition exposure offer significant information; for example, the mean or median exposure provide a broad view, but the exposure of the 90th or 95th percentile of consumers provides details on high consumers [37]

Considering the weight variances between adults and children, 20, 40, 62, and 60 kg were used to compute the exposure to Cd per kilogram of body weight and MOE values in the various age groups, 3~6, 7~17, 18~59, and ≥60 years, respectively. MOE values less than 1 signify health risk, in accordance with Health Canada’s MOE evaluation guidelines for genotoxic carcinogens [34].

### 2.7. Statistical Analysis

Probabilistic risk assessment model calculations for Cd dietary exposure, and MOE values were performed by using @RISK software (Palisade Corporation, 7.6. Industrial, 2018) based on a Monte Carlo simulation with 10,000 iterations. The results are expressed as means ± standard deviations to illustrate the concentration of Cd in foodstuffs, for which statistical analyses were performed using SPSS version 26. Value ranges are indicated by the 5th to 95th percentiles.

By employing advanced features available in the @RISK model, this study might provide (for the first time regarding risk estimation of Cd exposure) a comprehensive vantage point to facilitate a sophisticated analysis of simulation data, as the single point assessment technique used to be employed in this field [38,39,40,41,42], which does not permit all definitions of the risk concept as is required in the MOE model. The combination of the MOE method with @RISK analysis leads therefore to multiple point risk characterization, which aimed to improve the accuracy and comprehensiveness for the risk estimation of dietary intake of Cd in this study.

## 3. Results and Discussion

### 3.1. Cd Levels in Food

The levels of Cd in 3964 food samples were subdivided by food category and presented as detection rates, average values ± standard deviations, and 5th to 95th percentile ranges as presented in Table 1. The overall detection rate of Cd was 69.6% (1206/3964), with an average of 0.120 mg/kg, while the P_50_ and P_95_ was 0.010 mg/kg and 0.29 mg/kg, respectively. Cd was wholly detected in bivalves, shrimp and crab, root vegetable, and soy product samples (detection rate being 100%), and very high detection rates were noted in fresh fruits, steamed bread, leafy vegetables, pork offal, noodles, fresh beans, and bulb vegetables as 99.4, 99.2, 98.8, 98.5, 98.4, 96, and 95%. The average values of Cd content in bivalves, shrimp and bulb vegetables were 1.36 ± 1.86, 0.24 ± 0.49, and 0.19 ± 0.27 mg/kg, respectively, meaning that they ranked the top three among various types of foods. On the contrary, the lowest Cd values were detected in eggs (9.7%), beef and lamb (18.6%), poultry (38.4%), and edible fungus (60%). As described above, the Cd detection rates among different food categories collected from Guangzhou varied widely. Seafood sub-groups (bivalve, shrimp and crab) presented the highest detection rates, followed by bulb vegetables and rice, and rice products. Particularly, our results were similar to an assessment conducted on dietary exposure to Cd in adult residents from a total diet study in Shenzhen (a city near to Guangzhou) where high Cd detection rates in adult residents were found in vegetables, rice and its products, fish, seafood and shellfish [39].

### 3.2. Dietary Exposure of Cd

Notably, there is a historic variance in food cultures worldwide, particularly, between South and North China. Rice is hard to cultivate due to the drier Northern soils, and was historically expensive to transport from the South; hence, rice is not commonly consumed or preferred by residents in North China, while the South continues to produce rice. Populations in South China consume rice as staple food, whereas in the North, wheat flour (prepared as dumplings and noodles) is dominant. Meanwhile, research has revealed that Cd is principally accrued and dispersed in wheat roots and sprouts, but only an insignificant proportion of the metal can spread to the grains, hence wheat contributes little to Cd exposure in Northern populations. It therefore appears that regardless of the relatively high levels of Cd in certain foods (such as rice, rice products and sea foods), they may not contribute greatly to Cd EDI for those in North China, presumably due to their habitual low consumption of rice and seafood (except for the population residing along the east China seashore, such as in Dalian). There might be no detrimental effects of those foods on Cd-related health effects for residents in North China, but rather in the south (especially the Pearl River Delta) because, often, it may not necessarily represent the food with the highest Cd levels, but foods that are consumed in larger quantities which contain even moderate levels of Cd have the greatest impact on Cd dietary exposure [43].

The EDI of Cd from various foods in different age groups in Guangzhou is listed in Table 2. The food categories contributing most to Cd exposure were rice and rice products (8.63 ng/kg b.w./day), leafy vegetables (3.18 ng/kg b.w./day), bivalves (2.79 ng/kg b.w./day), and shrimp and crabs (1.48 ng/kg b.w./day). Of all the food categories, rice and rice products had the highest EDI value (8.63 ng/kg b.w./day), with the total dietary consumption of all age groups ranking the largest (135.8 ± 86.9 g/day), where an ascending tendency of the EDI value along with age advancement was noticed. This renders rice and rice products as the largest contributors towards Cd dietary exposure for Guangzhou citizens. Furthermore, it is of remarkable importance to note the relatively high total dietary consumption of Cd in leafy vegetables (122.2 ± 89.7 g/day), and the EDI (3.18 ng/kg b.w./day), both ranking second after rice and rice products. Similar to rice and rice products, leafy vegetables also showed an ascending trend with the advancement of age.

Our observation seems consistent with the findings of recent studies from some Chinese institutions that staple diets (rice, vegetables and seafood) remain the chief sources for dietary Cd exposure, while in another total diet study on contamination and health risk assessment of Cd in one of the Northern provinces of China, Jilin, the main sources of dietary exposure to Cd were vegetables and vegetable products [44]. Moreover, in a similar assessment performed in Liaoning Province (next to Jilin Province), the primary dietary sources of Cd included cereals, legumes, meat, aquatic products, and vegetables. Although in recent years, it has been discovered that fewer and fewer rice and rice products exceed the limit [45,46], the prevalence of the consumption of rice and its products as a staple diet by local residents in Guangzhou, represents a cause for concern because of the probable coincidence of the particularly high consumption of rice/rice products and increased susceptibility to Cd-induced health damage (due to genetic traits and/or other health conditions). Rice and rice products are also staple foods in some other countries in Asia. For example, the Cd exposure in Thailand demonstrates as similar tendency as the observation in this study; the major food groups that contribute the most to Cd exposure are rice and grains, shellfish and sea food, meat including edible offal, and vegetables [12].

In the study, bivalves also present a fairly high total dietary consumption of Cd (2.0 ± 8.4), and a relevant EDI (2.79 ng/kg b.w./day). Noteworthy too are the significant dietary consumption per reference person (DCRP) for shrimp and crab (5.7 ± 18.4 g/day), and EDI (1.48 ng/kg b.w./day). The dietary consumption of pork meat was the third highest (93 ± 67.4 g/day) of all food categories, however, determined by its low EDI level, it contributed to Cd exposure insignificantly. Moreover, the estimated EDI range of Cd for each age group is 12.5 ng/kg b.w./day to 24.1 ng/kg b.w./day, with a mean value of 21.82 ng/kg b.w./day and a 90% confidence interval extending from 0 to 2.0. Among all age groups, the ≥60 age group had the highest EDI (24.1 ng/kg b.w./day), while the 3~6 years populace had the lowest EDI (12.5 ng/kg b.w./day). The main source of Cd exposure in all age groups was rice, whose EDI level increased along with the increase in age.

Considering the considerable contribution of bivalves to dietary Cd exposure in this study, and the common availability of this type of seafood to Guangzhou residents, the involvement of bivalves in terms of their potential hazardous effects on human health (especially for those people who consume bivalves significantly more than average) is of concern. The presence of Cd in bivalves may be largely attributed to natural and anthropogenic sources [47], where bivalves have great capabilities to bioaccumulate various heavy metals including Cd from their sea environments hinging on genetic and geochemical factors [48,49]. A recent study suggested that ocean acidification has been intensifying Cd accrual in aquatic bivalves [50,51]. According to a study concerning the health risks of heavy metals on five chief marketed marine bivalves, sampled from three coastal cities in Guangxi province (Qinzhou, Fangchenggang, and Beihai), a southern-most coastal area of China, the Cd concentrations in eatable muscle of most bivalve samples were below the national and global safety limits. Nevertheless, even if the existing build-up levels of bivalves are generally safe, it is still possible that sustained and lifetime ingestion (e.g., beyond 70 years) becomes sufficient in posing adverse health effects [52]. Moreso, shrimp and crab are assumed to be capable of absorbing Cd in aquatic waters through respiratory and digestive systems and the external body without substantial excretion [28]. In contrast to our results, shrimp and crab commodities in some other regions of China, namely Shanghai, and Shanxi, Zhejiang, and Shandong provinces, obtained from either marine environment or internal rivers, presented an elevated risk of Cd dietary exposure, with its mean level surpassing the limit set by China [53]. This denotes possible health risks triggered by Cd, in both China and the rest of the world.

### 3.3. Risk Assessment and Characterization of Dietary Cd Exposure in Guangzhou Residents

The level of the MOE indicates the level of concern, and the larger the MOE, the lower the risk of the exposure. Table 3 shows the MOE values of Cd dietary exposure for populations of various age groups in Guangzhou. MOE values gradually increased along with the advancement of age; particularly, the 95% confidence distribution ranges for all age groups except for the 3~6-year age (youngest) group were above 1.0 (the marginal value for judging a health risk). In fact, the 95% confidence range of MOE for the total population was 2.29 (the 95% confidence interval extending from 1.92 to 4.81), and the MOE value of the 3~6-year age population was the lowest at 1.33, with a 95% confidence interval. The results may indicate that the 3~6-year age group is at a potential risk of dietary exposure to Cd.

In another relevant research work conducted in 2018 [42], the dietary exposure to Cd in residents of Guangzhou, China (food samples collected from 2013 to 2015) was assessed, and the risk characterization relied on (single) point assessment; therefore, only a conclusion about the whole population was made, with the specific calculation of the parameters (PTMI) in different age groups being impossible. On the contrary, in this study we used a multiple point assessment model to measure the MOE values in different age groups, leading to the identification of particular age group with a defined health risk. Due to the use of the MOE method in this study, our conclusion should prove to be more reliable. The MOE method is henceforth regarded as the most scientifically plausible and pragmatic approach to health risk characterisation, since it considers both dietary exposure and statistical evidence on the dose–response relationship, i.e., efficacy, without extending the evaluation beyond the observed dose-range or generating unclear risk estimations [54,55].

Concerning the MOE value of the 3~6-year age population being the lowest, i.e., 1.33 (0.92 to 3.12), Cd exposure and accumulation in the human body begins even at infantile age and secretion from the body is constrained. Its accumulation in the kidney is accountable for outcomes such as osteoporosis and nephrotoxicity in adulthood [56]. Cd exposure at young ages should be restricted to prevent direct influences on children and to inhibit potential health effects at an older age. Generally, children and infants may have higher exposure to metals as they consume more food than needed to only to maintain their body weight (in order to realise growth and development), and thereby they absorb metals more easily than adults [57].

### 3.4. Uncertainty Analysis

There are some uncertainties in this study. (1) We did not monitor the marine lives for their whole lifespan, or search for the origins of the exact food production sites, but presumed that most foods derived from the local province (areas within 300 km from Guangzhou). For example, seafood available in Guangzhou is ordinarily from the seashores nearest to the city. (2) We did not study the contents of Cd in the relevant soils. (3) Only 3-day food records were utilised to assess consumption data, and the short time frame may not reflect the eating habits of the whole year. Therefore, increasing the number of recoding days and repeating the consumption survey during different seasons may provide a more accurate assessment of habitual food consumption. Moreover, host features that can conceivably impact nutrient absorption could not be monitored in our study, hence there is a need for a further bio-accessibility and bioavailability targeted study.

## 4. Conclusions

Particularly in Guangzhou, a typical city in South China, the consumption of rice, rice products, and some sea foods contributes most to the daily intake of Cd. Taking the amounts of Cd ingested with all foods and the dietary structures of the local people at various age groups into account, the risk assessment of Cd intake using the MOE model and @Risk software indicated that the safety of dietary Cd exposure may gradually increase along with an elevation of age toward adulthood, and young children (3~6-year age) are uniquely under health risk to some extent, while people at more advanced ages are generally safe. Further studies and the protection of the local young children (particularly for cases with increased susceptibility to Cd poisoning) are encouraged.

## Figures and Tables

**Table 1 ijerph-19-09572-t001:** Levels of Cd in foods from Guangzhou during 2016–2019.

Food Category	Number of Samples	No. of Samples Below LOD	Detection Rate (%)	Level of Cd (mg/kg)			
				Means ± S.D.	P_50_	P_95_	Range
Rice and rice products	859	101	88.2	0.06 ± 0.08	0.04	0.19	ND~0.95
Leafy veggies	251	3	98.8	0.02 ± 0.46	0.01	0.06	ND~0.65
Bulb veggies	21	1	95	0.19 ± 0.27	0.01	0.99	ND~0.99
Root veggies	10	0	100	0.01 ± 0.01	0.01	0.03	ND~0.03
Fish	210	111	47.1	0.002 ± 0.005	0.01	0.01	ND~0.01
Shrimp and crab	170	0	100	0.24 ± 0.49	0.02	1.64	ND~2.36
Bivalve	260	0	100	1.36 ± 1.86	0.37	5.76	ND~10.7
Edible fungus	60	24	60	0.02± 0.43	0.002	0.16	ND~0.18
Pork offal	68	1	98.5	0.08 ± 0.12	0.03	0.28	ND~0.7
Pork	44	10	77.3	0.002 ± 0.004	0.001	0.01	ND~0.02
Beef and lamb	86	70	18.6	0.0005 ± 0.0008	0.01	0.11	ND~0.3
Vegetable oil	120	30	75	0.002 ± 0.002	0.002	0.006	ND~0.02
Egg/egg products	290	262	9.7	0.0004 ± 0.0003	0.01	0.001	ND~0.003
Fresh beans	50	2	96	0.004 ± 0.0003	0.007	0.01	ND~0.05
Fresh fruits	173	1	99.4	0.01 ± 0.02	0.004	0.06	ND~0.14
Maize	90	7	92.2	0.003 ± 0.003	0.001	0.009	ND~0.02
Melons	101	7	93.1	0.01 ± 0.01	0.005	0.036	ND~0.05
Noodles	563	9	98.4	0.01 ± 0.01	0.01	0.03	ND~0.05
Nuts	120	29	75.8	0.04 ± 0.07	0.01	0.2	ND~0.32
Poultry	177	109	38.4	0.0008 ± 0.001	0.01	0.003	ND~0.01
Soy products	110	0	100	0.01 ± 0.01	0.01	0.03	ND~0.03
Steamed bread	131	1	99.2	0.01 ± 0.01	0.01	0.02	ND~0.02
Total	3964	1206	69.6	0.12 ± 0.59	0.01	0.29	ND~10.7

ND: Not detected; LOD: Limit of detection; P50: the 50th percentile; P95: the 95th percentile.

**Table 2 ijerph-19-09572-t002:** The EDI of Cd from foods presumably consumed Guangzhou residents at varying ages.

Food Category	DCRP (g/day) in Groups at Varying Ages (Years)					EDI of Cd (ng/kg b.w./day) in Groups at Varying Age (Years)				
	3~6	7~17	18~59	≥60	Total	3~6	7~17	18~59	≥60	Total
Beef and lamb	7.6 ± 15.4	14 ± 23	16 ± 27	9.9 ± 17	15 ± 25	0.002 (0~0.04)	0.004 (0~0.03)	0.004 (0~0.05)	0.003 (0~0.04)	0.004 (0~0.04)
Bivalve	0.8 ± 4.2	1.9 ± 7.8	2.2 ± 8.9	2.2 ± 7.9	2.0 ± 8.4	1.12 (0~9.1)	2.65 (0~29.4)	3.07 (0~33.9)	3.07 (0~30.1)	2.79 (0~35.6)
Bulb vegetables	3.1 ± 9.5	4.5 ± 13	5.6 ± 14	6.9 ± 14	5.2 ± 13	0.09 (0~0.12)	0.13 (0~0.29)	0.17 (0~1.0)	0.21 (0~0.35)	0.16 (0~0.22)
Shrimp and crab	3 ± 8.6	4.9 ± 14	6.3 ± 20	6.7 ± 18	5.7 ± 18	0.78 (0~7.29)	1.27 (0~15.3)	1.63 (0~25.78)	1.74 (0~20.58)	1.48 (0~6.9)
Edible fungus	1.7 ± 7.4	2.5 ± 8.3	2.6 ± 9.6	4.3 ± 15.4	2.6 ± 9.6	0.73 (0~0.31)	0.11 (0~1.15)	0.11 (0~1.26)	0.18 (0~2.17)	0.11 (0~0.95)
Vegetable oil	0.1 ± 1.7	0 ± 0.3	0.2 ± 2.2	0.0005 ± 0	0.1 ± 1.8	0.0003 (0~0.01)	0.001 (0~0.01)	0.001 (0~0.01)	0 (0~0)	0.0003 (0~0.01)
Egg/egg products	34 ± 30	32 ± 30	31 ± 30	31 ± 26	31 ± 30	0.01 (0~0.03)	0.01 (0~0.08)	0.01 (0~0.04)	0.01 (0~0.04)	0.01 (0~0.06)
Fish	27 ± 29	38 ± 36	46 ± 43	49 ± 38	43 ± 41	0.008 (0~0.17)	0.01 (0~0.27)	0.01 (0~0.27)	0.02 (0~0.61)	0.01 (0~0.85)
Fresh beans	1.8 ± 5.4	4.1 ± 16	3.8 ± 14	2.2 ± 9	3.6 ± 14	0.11 (0~0.35)	0.19 (0~0.53)	0.19 (0~0.71)	0.16 (0~0.52)	0.18 (0~0.58)
Fresh fruits	45 ± 59	48 ± 72	47 ± 69	40 ± 58	47 ± 68	0.60 (0~1.59)	0.60(0~3.12)	3.23 (0~2.66)	0.49 (0~3.42)	0.58 (0~2.23)
Leafy veggies	64 ± 54	110 ± 75	132 ± 93	144 ± 98	122 ± 290	1.68 (0~5.62)	2.86 (0~6.37)	3.43 (0~12.20)	3.77 (0~10.5)	3.18 (0~8.18)
Maize	6.3 ± 8.8	7 ± 18	9.2 ± 28	15 ± 28	14 ± 842	0.19 (0~0.18)	0.03 (0~0.22)	1.04 (0~0.2)	0.04 (0~0.29)	0.03 (0~0.22)
Melons	48 ± 41	77 ± 65	96 ± 84	101 ± 102	89 ± 80	0.47 (0~2.33)	0.75 (0~3.10)	0.94 (0~4.18)	10.83 (0~4.70)	0.87 (0~2.61)
Noodles	33 ± 42	48 ± 59	52 ± 59	53 ± 62	50 ± 58	0.48 (0~1.58)	0.71 (0~2.51)	0.77 (0~2.30)	0.78 (0~3.12)	0.74 (0~2.08)
Nuts	2.5 ± 4.7	2.2 ± 7.6	2.2 ± 7.6	1.1 ± 3.9	2.2 ± 11	0.12 (0~0.89)	0.10 (0~1.22)	0.10 (0~0.69)	0.05 (0~0.33)	0.11 (0~1.03)
Pork	60 ± 43	88 ± 64	99 ± 70	94 ± 63	93 ± 67	0.32 (0~0.64)	0.47 (0~2.34)	0.52 (0~2.28)	0.50 (0~0.57)	0.49 (0~0.90)
Pork offal	2.7 ± 9.9	4.8 ± 14	5.7 ± 17	6.3 ± 17	5.3 ± 16	0.28 (0~1.43)	0.49 (0~3.62)	0.59(0~2.73)	0.64 (0~4.2)	0.54(0~2.93)
Poultry	28 ± 32	45 ± 41	50 ± 48	42 ± 37	47 ± 45	0.01 (0~0.101)	0.01 (0.01~0.11)	0.02 (0~0.12)	0.01 (0~0.11)	0.01 (0~0.14)
Rice and rice products	79 ± 46	121 ± 74	147 ± 91	148 ± 90	136 ± 87	4.99 (0 ~ 24.86)	7.71 (0 ~28.19)	9.31 (0~34.87)	9.44 (0~37.32)	8.63 (0~28.52)
Root veggies	15 ± 25	21 ± 35	22± 35	22 ± 37	26 ± 37	0.74 (0~2.55)	0.73 (0~3.43)	0.76 (0~3.18)	0.89 (0~9.12)	0.74 (0~2.71)
Soy products	18 ± 33	29 ± 50	29 ± 50	25 ± 42	28 ± 48	0.40 (0~2.11)	0.66 (0~3.21)	0.66 (0~2.55)	0.57 (0~2.34)	0.63 (0~3.45)
Steamed bread	40 ± 43	59 ± 65	49 ± 58	46 ± 54	50 ± 58	0.43 (0~1.23)	0.63 (0~1.62)	0.53 (0~1.62)	3.98 (0~2)	0.50 (0~1.67)
Total						12.5 (0~42.2)	20.1 (0~66.2)	23.4 (0~70.5)	24.1 (0~67.7)	21.8 (0~80.9)

EDI: Estimated daily intake, DCRP: dietary consumption per reference person. The EDI values of Cd are expressed as the 95% confidence intervals; the lower limits of intervals were negative, so they are expressed here as zero.

**Table 3 ijerph-19-09572-t003:** MOE-based risk characterization of dietary Cd exposure in Guangzhou residents at varying ages.

Age (Year)	Monthly Dietary Intake of Cd (μg/kg b.w.)	Contribution to PTMI (%)	MOE (95% Confidence Range)
3~6	18.8 (13.9~22.8)	75.0	1.33 (0.92~3.12)
7~17	15.1 (10.6~18.4)	60.3	1.65 (1.31~3.4)
18~59	11.3 (8.5~13.1)	45.4	2.21 (2.14~6.18)
>60	12.0 (10.7~15.9)	48.1	2.08 (1.5~4.93)
Total	10.9 (7.5~13.1)	43.6	2.29 (1.92~4.81)

Each MOE value was obtained by dividing PTMI (25 μg/kg B.W.) by EDI.

## Data Availability

Data are available upon readers’ request.

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
