# Peer review of "Probabilistic Risk Assessment of Dietary Exposure to Cadmium in Residents of Guangzhou, China—Young Children Potentially at a Health Risk"

_ijerph, 2022, doi:10.3390/ijerph19159572_

Round 1
Reviewer 1 Report
Comments and suggestions are provided in the attached file

Author Response
Many thanks to reviewer #1 for the comments and suggestions. Please see the uploaded responses.

Reviewer 2 Report
This work of “Probabilistic risk assessment of dietary exposure to cadmium in residents of Guangzhou, China ─ young children potentially at a health risk” presents interesting and useful subject of study. The manuscript was prepared briefly and logically, using passable English language. Some mistakes and shortcomings, which I found, do not diminish the value of presented research. All my comments are listed below.
1. What is the basis for setting concentration of Cd?
2. The manuscript should add more discussion on the results of different parts.
3. Please check the grammatical errors in the manuscript carefully.
4. Please check the format of the references to meet the requirements of IJERPH.
Author Response
We have tried to follow the reviewer's comments and suggestions.

Reviewer 3 Report
The paper titled "Probabilistic risk assessment of dietary exposure to Cadmium in residents of Guangzhou, China ─ young children potentially at a health risk" refers to a very important topic of health. The authors took care of the rich materials and methods by analyzing 3964 food samples collected from each of the 13 districts of Guangzhou between 2015 and 2019. As a researcher, I appreciate the multi-year experiments done in multiple localities because it allows for more accurate predictions in the future. As a researcher in statistical methods, what I miss is the examination of the interaction between localities and the years of research the so-called LxY interaction. In their study, the authors note differences between the northern and southern parts of China. The findings confirm that there is a strong relationship between diet and Cd values, but as if, for example, the study were supplemented with weather conditions, it is possible that it would turn out that in areas with lower rainfall, wheat in grain has less Cd. This is just a suggestion for consideration. I rate the work as valuable with very good conclusions. I encourage you to develop this interesting topic.
Author Response
WE thank the reviewer's comments and suggestion. For discussion about the influence of the frequency/volume of rainfall between north and south China on the absorption of Cd to crops and organisms, we may need a new study.

Reviewer 4 Report
The article submitted for review is interesting. It contains an important aspect concerning the problem of heavy metals (Cd) content in food caused by excessive environmental pollution. The abstract is described properly-, and contains a summary of the work. The introduction refers to the content of the work. The test methods are suitable for this type of research. Statistical elaboration is correct. Discussion and the presented results of proper research, based on the current literature. The conclusions are too short. It should be slightly expanded with the research results. I also propose to add a research hypothesis (not included in the text)The work with minor corrections can be published in the International Journal of Environmental Research and Public Health
Author Response
We completely agree with this reviewer's comments and suggestions. Many thanks to you!

Reviewer 5 Report
The present manuscript presents the results of testing for Cd-related health risk in the local Guangzhou residents at varying ages, by using a margin of exposure (MOE) model and RISK software. Practically, the authors made a statistic comparison between the estimated monthly exposures and the provisional tolerable monthly intake of Cd (based on Cd content in various food categories available locally).
Strength points:
-Originality/Novelty: The objective of the article is well defined and quite original, due to (MOE) model and RISK software.
-The introduction of the results provides sufficient background by including relevant references
-Interest to the Readers: The article has potential to attract a quite wide readership.
-Overall Merit: The article has potential to provide results of interest for both research and medical practice.
-English Level: The English language is appropriate and understandable.
Weakness points:
This article presents a quite limited approach on testing for Cd-related health risk in the local Guangzhou residents, since only 3-day food records were utilized to assess consumption data. Also, Cd bioavailability study is missing.
Author Response
We thank this reviewer for the comments and suggestion, which at least let us study more on the food recall investigation protocols and their validity.
